# HEPrune: Fast Private Training of Deep Neural Networks With Encrypted Data Pruning

Yancheng Zhang[1], Mengxin Zheng[1], Yuzhang Shang[2], Xun Chen[3], and Qian Lou[1]

[1]University of Central Florida
[2]Illinois Institute of Technology
[3]Samsung Research America
{yancheng.zhang,mengxin.zheng,qian.lou}@ucf.edu;
yshang4@hawk.iit.edu; xunchen@outlook.com

## Abstract

Non-interactive cryptographic computing, Fully Homomorphic Encryption (FHE), provides a promising solution for private neural network training on encrypted data. One challenge of FHE-based private training is its large computational overhead, especially the multiple rounds of forward and backward execution on each encrypted data sample. Considering the existence of largely redundant data samples, pruning them will significantly speed up the training, as proven in plain non-FHE training. Executing the data pruning of encrypted data on the server side is not trivial since the knowledge calculation of data pruning needs complex and expensive executions on encrypted data. There is a lack of FHE-based data pruning protocol for efficient, private training. In this paper, we propose, *HEPrune*, to construct a FHE data-pruning protocol and then design an FHE-friendly data-pruning algorithm under client-aided or non-client-aided settings, respectively. We also observed that data sample pruning may not always remove ciphertexts, leaving large empty slots and limiting the effects of data pruning. Thus, in HEPrune, we further propose ciphertext-wise pruning to reduce ciphertext computation numbers without hurting accuracy. Experimental results show that our work can achieve a $16\times$ speedup with only a $0.6\%$ accuracy drop over prior work. The code is publicly available at https://github.com/UCF-Lou-Lab-PET/Private-Data-Prune.

## 1 Introduction

Machine learning, especially deep neural networks, has been widely applied in various domains, including healthcare [1], finance [2], and so forth. Due to the lack of expertise and computational resources, the average user often outsources the training task to the cloud servers in the machine-learning-as-a-service (MLaaS) setting. However, the training data is often highly private and private, and it should not be directly shared because of business, legal, and ethical constraints. Private training enables the cloud server to train machine learning models on encrypted data, where strong privacy guarantees are offered by cryptographic primitives such as FHE [3].

However, FHE-based private training is significantly hindered by its substantial execution time. For example, the encrypted training can be $1 \sim 3$ orders of magnitudes slower than in the plaintext training [4, 5, 6]. The high execution time primarily stems from the transformation of a plaintext dataset into an encrypted format, along with the substitution of straightforward plaintext computations with more resource-intensive ciphertext operations. Reducing the number of data samples used for training could greatly accelerate the process. To optimize plaintext datasets, methods like dataset pruning are commonly employed, where the model trainer has access to both the dataset and model.

38th Conference on Neural Information Processing Systems (NeurIPS 2024).

Dataset pruning involves analyzing data samples, calculating an importance score for each sample during training iterations, and discarding redundant samples to streamline training. Prior research indicates that dataset pruning can enable machine learning model training on a fraction of the data—sometimes as little as 10%—with minimal to no loss in accuracy [7, 8, 9, 10, 11].

Pruning private datasets in encrypted form to accelerate training remains an open problem due to several challenges. First, calculating importance scores and sorting data samples for removal require complex encrypted computations, such as costly non-linear and sorting operations. For instance, calculating the importance score requires tracking changes in training dynamics [11, 12]—such as logits, gradients, and losses—and computing corresponding distances on encrypted data, all of which involve costly FHE operations. These high overheads that may offset—or even surpass—the potential benefits of dataset pruning. Second, plaintext-level data pruning used in the prior plaintext data pruning [7, 8, 9, 10, 11] does not always yield execution and ciphertext reduction benefits, as a single ciphertext often contains multiple plaintext samples within its slots. Therefore, effective ciphertext pruning is critical to enable faster private training.

In this paper, we propose *HEPrune*, the first encrypted data pruning framework for private training. We begin by constructing an encrypted data pruning method using pure FHE. To accelerate FHE-enabled data pruning, we introduce a homomorphic encryption-friendly score (HEFS) to effectively evaluate the importance of data samples. To avoid costly FHE-based sorting, we propose client-aided masking (CAM) to identify less important samples while ensuring data privacy. To bridge the gap between sample-wise importance scores and ciphertext-wise data, we develop a ciphertext-wise pruning (CWP) technique to reduce the number of ciphertexts during private training. We conduct experiments in both training-from-scratch and transfer-learning settings. The proposed encrypted data pruning framework can accelerate private training by $16\times$ with only a $0.6\%$ accuracy drop.

## 2 Background

### 2.1 Fully Homomorphic Encryption-Enabled Private Training

Fully Homomorphic Encryption (FHE)[3] is a cryptographic primitive that supports arbitrary computation on encrypted data. An FHE workflow involves four functions: client-side `KeyGen`, `Enc`, `Dec`, and server-side `Eval`. For `Eval`, FHE natively supports homomorphic addition, subtraction, and multiplication between ciphertexts and plaintexts, denoted as $\boxplus$, $\boxminus$, and $\boxtimes$, respectively. Non-linear functions such as the comparison function, max function, SoftMax function, and square root function can be approximated using polynomials. We denote these functions as HE.Cmp[13, 14], HE.Max[14], HE.SoftMax[6, 15], and HE.Sqrt[16], respectively. FHE enables confidential training on encrypted datasets. While early works focused on training simple models on small datasets, such as linear regression and logistic regression models[17, 18, 19, 20], recent research has advanced towards training deep neural networks (DNNs) using HE. Although HE provides theoretical data privacy guarantees, privately training DNNs on large encrypted datasets typically incurs substantial computational overhead. For example, training a dense model for one step on a batch of $60$ MNIST samples can take over $1.5$ days in FHESGD [4].

A line of research has been proposed to accelerate HE-enabled private training. Glyph [5] incorporates two HE schemes, TFHE and BGV, to enable private training in the transfer learning setting, allowing training of an MNIST mini-batch in only $0.04$ hours. The most recent work, HETAL [6], achieves private training within 1 hour on the MNIST and CIFAR-10 datasets by using optimized matrix multiplication and GPU acceleration. While other cryptographic tools, such as Multi-Party Computation (MPC), can also achieve private training, most existing MPC approaches [21, 22, 23, 24] rely on a non-colluding server assumption, where multiple non-colluding servers are required to ensure security. Additionally, MPC-based methods typically incur significant communication overhead; training a LeNet model on the MNIST dataset, for instance, can generate approximately $500\,GB$ of communication overhead [24]. In contrast, FHE-enabled private training does not require the non-collusion assumption and incurs significantly less communication overhead.

### 2.2 Dataset Redundancy and Pruning

Not all data samples contribute equally to DNN training [25]; some samples are less important and can be pruned during training. Training on only a subset of data can effectively reduce computational

overhead while achieving generalization performance comparable to training on the full dataset [10, 11]. Several methods have been proposed to identify the most informative data samples. Among these, score-based methods [7, 8, 9, 10] are widely used for their simplicity and effectiveness. These methods compute sample-wise importance scores during training and select the most informative samples based on these scores.

The importance score can be as simple as the entropy loss [8, 10] for each sample. Another commonly used score is the forgetting score [7], which counts the number of "forgetting events" that occur for each sample during training, where a forgetting event is defined as a change in the model's prediction for the same sample between two consecutive epochs. GraNd [9] uses the $\ell_2$-norm of the sample-wise gradient to represent sample importance, while ELN2 [9] approximates GraNd by calculating the $\ell_2$-norm of the error vector. Additional methods for identifying sample importance include submodularity-based methods like GraphCut [26] and gradient-matching-based methods like GradMatch [12]. These methods typically start with an empty set and gradually add samples, in contrast to score-based methods, where sample importance scores are computed simultaneously and independently.

Data pruning methods are generally divided into static and dynamic pruning methods. Static pruning refers to data pruning performed once before training [7, 8, 9, 26], while dynamic pruning involves pruning data every few epochs during training [12, 11, 10]. Although these methods were designed with different motivations and target settings, they can be adapted for use in either setting. For instance, [8] uses entropy loss in a static setting, while [11, 10] apply it in a dynamic setting. Similarly, dynamic pruning methods like [26, 12] can be adapted to static settings, as demonstrated by [27], without losing effectiveness. In this paper, we focus on dynamic pruning.

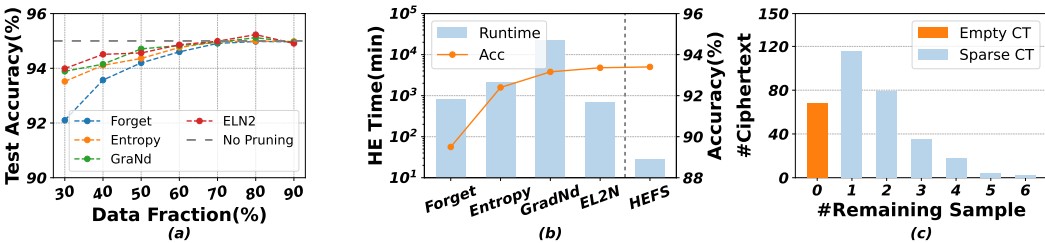

Figure 1: (a). Test accuracy under various pruning fractions in plaintext dataset (ResNet-18 on CIFAR-10). (b) The encrypted training overhead and accuracy of different data pruning methods. (c) The distribution of data sample numbers in a ciphertext after data pruning. Only a small fraction of ciphertexts are empty with a naïve sample-wise pruning.

**Motivation.** FHE-enabled private training provides a strong privacy guarantee for user data. However, private training remains significantly slower in runtime compared to unencrypted training. One primary reason for this is the large dataset size typically used for training, often comprising tens of thousands of samples. As shown in Figure 1(a), removing a fraction of data during training has minimal impact on model accuracy: even with only $30\%$ of the data, the accuracy drop is within $2\%$. However, directly applying plaintext data pruning methods to private training might not only fail to accelerate the process but could actually extend the training time. As shown in Figure 1(b), computing importance scores with existing methods—such as the forgetting score [7], entropy score [11, 10], GraNd [9], and EL2N [9]—can introduce prohibitive overhead. This inefficiency arises because FHE natively supports only addition and multiplication, while existing data pruning methods often require a range of non-linear operations. Although these non-linear operations can be implemented in FHE through lookup tables or approximations, these computations are extremely costly in FHE, potentially negating the benefits of data pruning. Furthermore, as illustrated in Figure 1(c), although existing methods can effectively prune samples from datasets, they do not necessarily reduce the number of ciphertexts in private training. This is because, during private training, a single ciphertext can contain multiple samples. Naïvely applying sample-wise data pruning can lead to a large number of sparse ciphertexts that cannot be excluded from training.

**Threat Model.** We consider a private training scenario where a client outsources model training to a cloud server. Our protocol is designed to allow a client to outsource model training while preserving the privacy of their data and model weights. The client's private dataset and model weights are treated as intellectual property that must not be disclosed to the server at any point. We assume the server is

semi-honest, meaning it will follow the protocol specifications but may attempt to gain unauthorized knowledge, such as the dataset and model weights. Sharing certain meta-information about training, such as the model architecture, dataset size, and early stopping signal, is assumed to be safe [6]. Side-channel attacks are beyond the scope of this paper.

# 3  HEPrune Design

We begin by outlining the pipeline for private training with encrypted data pruning in Section 3.1. In Section 3.2, we provide details of our FHE-based data pruning algorithm. Specifically, we construct an encrypted data pruning baseline to demonstrate how sample importance evaluation and sample removal can be performed using pure FHE operations. We then enhance the efficiency of the data pruning algorithm through FHE-friendly scoring (HEFS) and client-aided masking (CAM). Finally, in Section 3.3, we introduce ciphertext-wise pruning (CWP), which removes sparse ciphertexts and further accelerates the data pruning process.

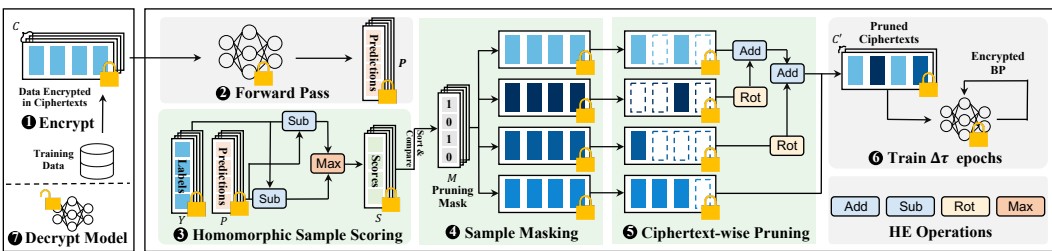

Figure 2: The overall workflow of private training with encrypted data pruning.

## 3.1  Pipeline of Private Training with Encrypted Data Pruning

It has been widely studied how to perform the gradient descent algorithm in the encrypted state during FHE-enabled private training [4, 5, 6]. However, it remains unclear how data pruning can be incorporated into existing private training frameworks. We illustrate our FHE-based encrypted data pruning framework in Figure 2. During private training, ❶ the client first encrypts the dataset with FHE and sends the encrypted dataset to the server. From this point forward, the dataset will never be decrypted. We refer to the encrypted data samples as ciphertexts. To identify the most informative subset of samples, ❷, the server first runs the forward pass algorithm on all ciphertexts to obtain the encrypted prediction vector $P$. ❸ With $P$ and the encrypted ground truth label $Y$, the server computes the sample-wise importance score $S = \mathcal{H}(x)$. We propose an HE-friendly importance score in Section 3.2. ❹ With the importance score, the server can decide which samples to prune. This is achieved by sorting the importance score and finding the threshold importance score $\mathcal{H}$ corresponding to the pruning ratio [7]. A pruning mask is generated by comparing the importance score with the threshold as $M = \mathbb{1}\{\mathcal{H}(x) < \mathcal{H}_t\}$. The unimportant samples can be masked by simply multiplying the ciphertexts with the corresponding masks. ❺ To fully reduce the number of ciphertexts, the server performs ciphertext-wise pruning according to the current pruning mask. ❻ The server performs a backpropagation algorithm on the pruned ciphertexts for $\Delta\tau$ epochs, and then ❷ starts a new round of the data pruning algorithm to update the training subset. ❼ At the end of private training, the server sends the encrypted model back to the client, and the client decrypts the model.

We formalize the above workflow into an encrypted data pruning protocol in Figure 5 in Appendix A. Our protocol is the first of its kind to introduce data pruning to private training and is an enhanced private training framework with encrypted data pruning extension over existing works [4, 5, 6]. Additionally, our protocol can work in both the transfer learning setting [5, 6] and training-from-scratch setting [4]. Our protocol is fully compatible with the early stop techniques adopted in [6]. As in [6], sending meta information during training is allowed in our protocol, such as the logits, early stop signals, and importance scores. The privacy of data and models is strongly guaranteed as they are encrypted and never decrypted by the server. We first instantiate a baseline of encrypted data pruning using only FHE operations in Section 3.2, and then propose HEFS and CAM to optimize the overhead of the encrypted data pruning algorithm. In Section 3.3, we propose CWP to effectively reduce the number of ciphertexts involved in training and thus boost the efficiency of private training.

## 3.2 HE-enabled Data Pruning

While FHE can support arbitrary computation on encrypted data, computing non-linear functions is typically expensive. Applying existing data pruning methods to private training can lead to additional overhead that negates or even exceeds the benefits of performing data pruning. For the entropy-based methods [8, 11, 10], although the entropy loss is typically considered cost-free in the plaintext, it is not explicitly computed during private training. Instead, we compute the gradient directly without calculating the loss itself, as illustrated in Figure 5, step 3(b). Computing the entropy loss in FHE requires approximating the logarithm function, which is computationally expensive. The forgetting score [7] is simpler to compute, requiring only comparison operations between the current prediction and prediction from the last epoch. However, the forgetting score cannot be easily made dynamic, as it usually requires calculations over the full dataset for multiple epochs. The most viable existing data pruning method is the EL2N [9], which simply utilizes the prediction $P$ and label $Y$. Although originally proposed as a static pruning method, we find that EL2N also remains effective in a dynamic setting, similar to the entropy score [10]. We first instantiate an FHE-enabled data pruning algorithm, detailed in Algorithm 1, based on the dynamic version of EL2N (HE2LN). Subsequently, we demonstrate that even EL2N remains computationally expensive in the encrypted state and propose the HE-friendly score (HEFS) and client-aided masking (CAM) to accelerate the encrypted data pruning.

---

**Algorithm 1:** Homomorphic Data Pruning

**Input** : The encrypted dataset $\bar{D} = \{\bar{X}_i, \bar{Y}_i\}_{i=0}^{C-1}$ and the model $\mathcal{M}$ with weights $\bar{W}$, the pruning ration $p$, the sample number $N$ and the target class number $n$.

**Output** : The pruned encrypted dataset $\bar{D}' = \{\bar{X}'_i, \bar{Y}'_i\}_{i=0}^{C'-1}$, where $C' \leq C$.

**for** $i \leftarrow 0$ **to** $C-1$ **do**
   $\bar{P}_i \leftarrow$ HE.SoftMax($\mathcal{M}(\bar{X}_i; \bar{W})$) ;                   // Prune.Eval
   $\bar{err}_i \leftarrow \bar{P}_i \boxminus \bar{Y}_i$ ;                   // compute the error vector
   $\bar{err}_i \leftarrow \bar{err}_i \boxtimes \bar{err}_i$;
   $err\_\bar{s}um_i = \bar{err}_i$;
   **for** $j \leftarrow 0$ **to** $\log n - 1$ **do**
      $\bar{err}\_t_i \leftarrow$ HE.Rotate($err\_\bar{s}um_i, 2^j$);
      $err\_\bar{s}um_i \leftarrow err\_\bar{s}um_i \boxplus \bar{err}\_t_i$;
   $\bar{score}_i \leftarrow$ HE.Sqrt($err\_\bar{s}um_i$) ;          // compute the importance score

$s\_\bar{score} \leftarrow \{\}$;                            // Prune.Remove
$B \leftarrow N/C$;
**for** $i \leftarrow 0$ **to** $C-1$ **do**
   **for** $j \leftarrow 0$ **to** $B-1$ **do**
      $s\_\bar{score}_{iC+j} \leftarrow$ HE.Rotate($\bar{score}_i, j$) ;      // extract sample-wise scores

**for** $i \leftarrow 0$ **to** $N-1$ **do**
   **for** $j \leftarrow 0$ **to** $N-i$ **do**
      $\bar{less} \leftarrow$ HE.Cmp($s\_\bar{score}_i, s\_\bar{score}_j$);       // sort the sample-wise scores
      $(s\_\bar{score}_i) \leftarrow (\bar{less} \boxtimes s\_score_j) \boxplus ((1 \boxminus \bar{less}) \boxtimes s\_score_i)$;
      $(s\_\bar{score}_j) \leftarrow (\bar{less} \boxtimes s\_score_i) \boxplus ((1 \boxminus \bar{less}) \boxtimes s\_score_j)$;

$threshold\_index \leftarrow N \times p$;
$\bar{\theta} \leftarrow s\_\bar{score}_{threshold\_index}$;
**for** $i \leftarrow 0$ **to** $C-1$ **do**
   $\bar{mask} \leftarrow$ HE.Cmp($\bar{score}_i, \bar{\theta}$);             // remove unimportant samples
   $(\bar{X}_i, \bar{Y}_i) \leftarrow (\bar{mask} \boxtimes 0) \boxplus ((1 \boxminus \bar{mask}) \boxtimes (\bar{X}_i, \bar{Y}_i))$;
**return** $\bar{D}' \leftarrow \{\bar{X}'_i, \bar{Y}'_i\}_{i=0}^{C-1}$

---

**HEL2N Baseline.** The EL2N score is defined as $\mathcal{H}(x) = \mathbb{E}_{w_t} \|p(x; w_t) - y\|_2$, where $p(x; w_t)$ is the prediction vector for sample $x$ and $y$ the corresponding ground truth label. In essence, the EL2N score is the $\ell_2$-norm of the error vector, which can be computed via pure FHE operation as shown in Algorithm 1. We first compute the prediction vector $P$ by encrypted forward pass. The HE.SoftMax in the forward pass is evaluated by polynomial approximation [15] and domain extension technique [28]. Computing the forward pass in the encrypted state is the same as existing private training frameworks [6]. With the encrypted prediction vector $P$ and the ground truth label

$Y$, we can compute the EL2N score homomorphically. Specifically, we first compute the sum of squares over the error vector, which requires homomorphic subtraction, multiplication, and rotation. Then we take the square root of sum of squares to compute the $\ell_2$-norm of the error vector. While HE.Sqrt can be implemented via Newton iterative algorithm [16], the overhead is prohibitive as the Newton iterative algorithm needs $\sim 10$ iterations to compute an accurate square root. Additionally, the Newton iterative algorithm in FHE involves considerable ciphertext-ciphertext multiplication. This leads to a large number of additional relinearization and bootstrapping operations. A single square root over a ciphertext can take up to 2 minutes.

**HE-friendly Score.** Simple as the EL2N score is, using it to evaluate the importance of data samples in the encrypted state can still incur prohibitive overhead due to its complex non-linearity. We propose an HE-friendly importance score (HEFS) to address this issue. We first derive the formulation of HEFS, and then demonstrate how it can be computed via pure FHE operation. To determine how a single data sample affects the training, we can quantify the importance of a sample by the difference of the gradient before and after removing a sample. Denote the gradient of a sample $(x, y)$ over the weights at time $t$ as $g_t(x, y) = \nabla_{w_t}\ell(p(w_{t-1}, x), y)$. Given a minibatch of $B$ samples $S = \{(x_i, y_i)\}_{i=0}^{B-1}$, the importance of a sample can be quantified by the difference of the time derivative of the loss function, $\Delta_t$, before and after removing the sample from the minibatch. The difference of the derivative is bounded by the sample's gradient [9]. Specifically, let $S_{\neg k} = S \setminus \{(x_k, y_k)\}$ be the set after removing a certain sample $(x_k, y_k)$. For $\forall(x_i, y_i) \in S$ and $i \neq k$, it holds that

$$\|\Delta_t((x_i, y_i), S) - \Delta_t((x_i, y_i), S_{\neg k})\|_2 \leq c\|g_t(x_k, y_k)\|_2 \tag{1}$$

EL2N approximates the $\ell_2$-norm of the gradient by the $\ell_2$-norm of the error vector. We further streamline the EL2N score using the $\ell_1$-norm. More formally, we define the HEFS as:

$$\mathcal{H}(x) = \mathbb{E}_{w_t}\|p(x; w_t) - y\|_1 \tag{2}$$

HEFS can be efficiently computed during private training. Specifically, the circuit for HEFS consists of only two FHE subtractions and one max operation, which can be computed as:

$$score = \mathsf{HE.Max}((Y \boxminus P), (P \boxminus Y)) \tag{3}$$

In the above equation, the homomorphic subtraction $\boxminus$ is significantly faster than other FHE operations. HE.Max can be effectively computed via HE.Cmp. While the HE.Max operation has the same time complexity as HE.Cmp, the concrete runtime of HE.Max is even more efficient, typically $1.5 \sim 2\times$ faster under the same parameter setting [14]. The proposed HEFS is a close approximation to the original EL2N score, which guarantees the effectiveness of the HEFS-based data pruning. We show the accuracy of the data pruning using HEFS in Section 4.

---

**Algorithm 2:** Client-aided Sample Pruning

**Input**  : The pruning ratio $p$. The server holds the encrypted dataset $\bar{D} = \{\bar{X}_i, \bar{Y}_i\}_{i=0}^{C-1}$ and encrypted score $\overline{score}$ and the client holds the secret key SK.

**Output** : The pruned encrypted dataset $\bar{D}' = \{\bar{X}'_i, \bar{Y}'_i\}_{i=0}^{C'-1}$, where $C' \leq C$.

**Client:**
> $score \leftarrow \mathsf{HE.Dec}(\overline{score}, \mathsf{SK})$;
> $\theta \leftarrow \mathsf{QuickSelect}(score, p)$;
> $mask \leftarrow \mathsf{Compare}(score, \theta)$;
> Sends $mask$ to the server;

**Server:**
> $\bar{D}' \leftarrow \bar{D}$;
> **for** $i \leftarrow 0$ **to** $C - 1$ **do**
> > **if** $mask_i == 0$ **then**
> > > $\bar{D}' \leftarrow \bar{D}' \setminus \{(\bar{X}_i, \bar{Y}_i)\}$; // remove an empty ciphertext
> >
> > **else**
> > > $(\bar{X}_i, \bar{Y}_i) \leftarrow mask_i \boxtimes (\bar{X}_i, \bar{Y}_i)$; // remove unimportant samples

**return** $\bar{D}'$

---

**Client-aided Masking.** After evaluating the importance of each data sample, we need to remove the less informative ones from the training set. This requires the server to sort all importance scores homomorphically to determine the threshold of important scores, $\bar{\mathcal{H}}$. Given a dataset with $N$ data samples, $O(N^2)$ homomorphic comparisons are needed. Since $N$ is typically large for machine learning model training, such sorting incurs prohibitive overhead in the encrypted state, which can offset the benefits of data pruning or even prolong the total training time. To effectively identify and remove the less important data samples, we propose client-aided masking (CAM) in Algorithm 2. We offer an analysis on the computation, communication overhead, and security implications as follows.

**Efficiency.** In contrast to the heavy server-side homomorphic sorting, finding $\mathcal{H}_t$ is extremely fast on the client's side, with only $O(N)$ runtime via the quick select algorithm. In practice, this process takes only $15\ ms$ on the CIFAR-10 dataset. Additionally, the communication overhead is minimal.

For the CIFAR-10 dataset with $43,750$ training samples, only 2 CKKS ciphertexts are needed to transfer the encrypted importance scores when the slots are fully utilized. This incurs only 4 MB of communication overhead. For comparison, the early stopping signals used to determine early stopping incur 18 MB communication overhead when transferring the encrypted logits [6].

**Security.** Directly asking the client to decrypt and reveal the model weights and datasets to the server clearly breaches the client's privacy and is therefore prohibited. However, exchanging meta information about training does not directly compromise private information and is typically permitted. For instance, HETAL [6] allows the server to send the logits of the validation set to the client, who then computes the loss and determines whether to stop training early. Such exchanges of meta information are crucial to ensure the effectiveness of private training. Further details can be found in Appendix B.

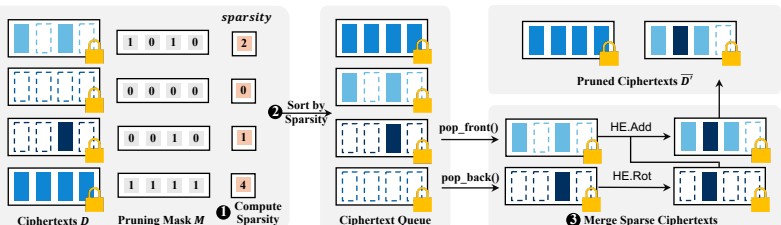

Figure 3: Example of ciphertext-wise pruning.

### 3.3 Ciphertext-wise Pruning

While the HEFS and CAM make the encrypted data pruning algorithm much more practical, the resulting ciphertexts remain largely sparse, thus limiting the training time acceleration. To this end, we propose ciphertext-wise pruning to effectively reduce the number of ciphertexts involved in training, as illustrated in Figure 3, which further boosts the efficiency of the private training. We detail the steps of ciphertext-wise pruning in Algorithm 3. Without loss of generality, we refer to all the slots occupied by a single sample within a ciphertext as a sample slot. Once the server obtains the pruning mask, ciphertext-wise pruning can be performed without client involvement.

We denote the number of samples in each ciphertext as $B$ and $M_i$ is a Boolean array that indicates whether each of the $B$ samples should be pruned. ❶ The server first computes the sparsity of each ciphertext, which is done by simply counting the number of 0-s in each $M_i$. ❷ After computing the ciphertext-wise sparsity, the server can sort the ciphertexts along with their corresponding masks. We represent the sorted ciphertexts in a dequeue. To perform ciphertext-wise pruning, the server first identifies two ciphertexts $ct_{front}$ and $ct_{back}$ from the queue. ❸ After removing the full ciphertexts and empty ciphertexts from the queue via $\mathsf{Trim}()$, we then leverage the most sparse ciphertext $ct_{back}$ to fill

---

**Algorithm 3:** Ciphertext-wise Pruning

**Input** : The encrypted dataset
$\bar{D} = \{\bar{X}_i, \bar{Y}_i\}_{i=0}^{C-1}$ and pruning masks
$mask = \{M_i\}_{i=0}^{C-1}$, where
$M_i \in \{0, 1\}^B$ and $B$ is the number of
samples in each ciphertexts.

**Output** : The pruned encrypted dataset
$\bar{D}' = \{\bar{X}'_i, \bar{Y}'_i\}_{i=0}^{C'-1}$, where $C' \leq C$.

$\bar{D}' \leftarrow \emptyset$;
**for** $i \leftarrow 0$ **to** $C-1$ **do**
    $spasity_i \leftarrow B - \sum_{j=0}^{B-1} M_i[j]$; // compute
    the ciphertext-wise sparsity

$ct\_queue \leftarrow Sort(spasity, \bar{D}, mask)$;
$ct\_queue \leftarrow \mathsf{Trim}(ct\_queue)$;
**while** $ct\_queue.is\_not\_empty()$ **do**
    $ct_{front} \leftarrow ct\_queue.pop\_front()$;
    $ct_{back} \leftarrow ct\_queue.pop\_back()$;
    $slot_{empty} \leftarrow \mathsf{FZero}(ct_{front})$;
    $slot_{used} \leftarrow \mathsf{FOne}(ct_{back})$;
    $k \leftarrow slot_{used} - slot_{empty}$;
    $ct_{align} \leftarrow \mathsf{Mask}(ct_{back})$;
    $ct_{front} \leftarrow ct_{front} \boxplus \mathsf{HE.Rot}(ct_{align}, k)$;
    **if** $ct\_queue.is\_empty() \vee$
    $ct_{front}.sparsity == 0$ **then**
        $\bar{D}' \leftarrow \bar{D}' \cup \{ct_{front}.ct\}$;
    **else**
        $ct\_queue.push\_front(ct_{front})$;
    $ct\_queue.push\_back(ct_{back})$;

**return** $\bar{D}'$

---

in the empty slots in the least sparse ciphertexts $ct_{front}$. The key step is to align samples in $ct_{back}$ with the empty slots in $ct_{front}$. The function $\mathsf{FZero}()$ returns the index of the first empty sample slot $slot_{empty}$ in $ct_{front}$, sets the corresponding mask to 1, and decreases the sparsity of $ct_{front}$ by 1. Similarly, $\mathsf{FOne}()$ returns the first non-empty sample slot $slot_{used}$ in $ct_{back}$ and sets the corresponding mask and sparsity. By taking the difference of $slot_{used}$ and $slot_{empty}$, the server can

determine how much the $ct_{back}$ should be rotated to align with $ct_{front}$. We mask out the other slots in $ct_{back}$ and obtain $ct_{align}$ and keep only the first sample slot $slot_{used}$. This ensures only the empty sample slot in $ct_{front}$ is filled and the non-empty slots in $ct_{front}$ will not be corrupted. Then, $ct_{align}$ is rotated and added to $ct_{front}$ to fill in the empty slot in $ct_{front}$. After merging the two ciphertexts, if $ct_{front}$ is full or there are no remaining ciphertexts to merge, the server adds it to the final set $\bar{D}'$. We defer the details about Trim(), FZero() ,Fone() and Mask() to the Appendix D. We remark the pruning mask generated by CAM is not encrypted, while enables efficient operations related to the pruning mask without invoking heavy FHE computation.

## 4 Experiments

### 4.1 Experimental Setup

**Models and Datasets.** To demonstrate the generalizability, we evaluate the proposed encrypted data pruning methods in two settings, the transfer learning setting as in HETAL [6] and training-from-scratch setting as in FHESGD [4]. For the transfer learning setting, we adopt the same feature extractors as [6], the pre-trained ViT Base [29] model and the pre-trained MPNet Base [30] model. The client first extracts samples into a 768-dimension feature and then encrypts the features. The server performs private training on the encrypted features. For the training-from-scratch setting, the client encrypts the raw dataset directly, and the server performs private training on the encrypted dataset. We use a 3-layer MLP, with two hidden layers of dimension 128, which is a widely used structure in the private training setting [4, 21]. We perform encrypted data pruning on four widely used image datasets: MNIST [31], CIFAR-10 [32], Face Mask Detection [33], DermaMNIST [34] for image classification and one audio dataset SNIPS [35] for sentiment analysis. We partition the datasets as training, validation, and test set. The size of the training set and validation set is $7 : 1$. We defer the number of samples in each set in the Appendix C.

**System Setup and Implementation.** We use the RNS version of the CKKS [36, 37] scheme for homomorphic encryption. We use the bootstrapping method for CKKS in [38]. Our implementation is based on the OpenFHE [39] library. For HE parameters, we set the cyclotomic ring dimension as $N = 2^{16}$ and ciphertext modulus $1555$ bits to guarantee a security level of 128-bit under the Homomorphic Encryption Standard [40]. Each ciphertext has $N/2 = 32768$ slots and we set the multiplicative depth as $L = 12$. All experiments were conducted using an AMD Ryzen Threadripper PRO 3955WX processor operating at 2.2GHz, equipped with 125GB of memory. We use the Single-Instruction-MultipleData (SIMD) technique [41] to fully utilize the ciphertext slots and amortize the cost of homomorphic operations. Under SIMD, multiple data samples can be coded into one ciphertext. We adopt the most efficient encoding methods proposed in [6, 42]. For nonlinear operations like SoftMax and ReLU, we adopt approximation-based methods. We report the CPU version of HETAL, as their GPU implementation is not publicly available.

### 4.2 Results

**End-to-end performance.** As shown in Table 1, we compare the end-to-end training time and accuracy on five datasets with HETAL [6] and an unencrypted baseline. HETAL is known to be the most efficient full-data private training framework. For fair comparison, we unify the batch size as 128. For HETAL, we maintain the security parameters used in their original paper. For the proposed encrypted data pruning method, we set the pruning ratio as $p = 0.9$, i.e., only $10\%$ of the data remains for training in each epoch. The total training time is reduced by $\sim 6.6\times$ across datasets. The accuracy drop is as small as $0.25\%$ on the Face Mask Detection dataset. With encrypted data pruning, the accuracy is even $0.14\%$ higher than the unencrypted baseline and HETAL.

Table 1: End-to-end comparison across different datasets The pruning ratio is set as $p = 0.9$.

| Method | | MNIST | CIFAR-10 | Face Mask Detection | DermaMNIST | SNIPS |
|---|---|---|---|---|---|---|
| Unencrypted | Acc(%) | $95.69_{\pm 0.02}$ | $96.62_{\pm 0.02}$ | $95.46_{\pm 0.06}$ | $75.91_{\pm 0.11}$ | $94.43_{\pm 0.05}$ |
| HETAL | Acc(%) | $96.27_{\pm 0.02}$ | $96.57_{\pm 0.04}$ | $95.46_{\pm 0.05}$ | $76.06_{\pm 0.18}$ | $95_{\pm 0.08}$ |
| | Runtime(h) | 276.75 | 293.3 | 32.88 | 101.55 | 113.7 |
| Ours | Acc(%) | $95.54_{\pm 0.05}$ | $96.31_{\pm 0.06}$ | $95.21_{\pm 0.06}$ | $75.86_{\pm 0.15}$ | $95.14_{\pm 0.08}$ |
| | Runtime(h) | 41.89 | 44.76 | 5.02 | 15.5 | 17.36 |

Table 2: Effectiveness of the proposed method. We tested different private training methods on the CIFAR-10 dataset, with a pruning ratio $p = 0.9$. Communication is the size of logits and importance score ciphertexts server sends to client for early stopping and data pruning.

| Method | Accuracy(%) | Runtime(h) | Speedup | Communication(MB) |
|---|---|---|---|---|
| Full Data(HETAL) | $96.57_{\pm 0.04}$ | 293.3 | $\times 1$ | 18.1 |
| Prune Baseline | $95.98_{\pm 0.12}$ | 488.91 | $\times 0.6$ | 18.1 |
| +Client Aided | $96.16_{\pm 0.07}$ | 196.91 | $\times 1.49$ | 22 |
| +HEFS | $96.31_{\pm 0.06}$ | 105.57 | $\times 2.78$ | 22 |
| +Ciphertext-wise Pruning | $96.31_{\pm 0.06}$ | 44.76 | $\times 6.55$ | 22 |

**Effectiveness of the proposed methods.** In Table2, we demonstrate the effectiveness of each proposed technique. We start by training on the full CIFAR-10 dataset using the framework of HETAL. For the encrypted data pruning methods, we fix the pruning ratio as $p = 0.9$. In the prune baseline, we naively apply plaintext data pruning method to HETAL, which prolongs the total training time. Sorting 43750 importance score in CIFAR-10 can take more than 200 hours, which offsets the benefits of data pruning. By incorporating the client-aided masking, the server can remove the unimportant samples more effectively. The client-aided methods speed up the private training by $1.49\times$ with a $0.41\%$ accuracy drop. The communication cost increases slightly by approximately $1.2\times$. Yet, as the EL2N score involves costly square root computation, the speed-up is modest. By incorporating HEFS, we improve the overhead during importance score computation. With HEFS, the private training can be accelerated by $2.78\times$. The HEFS also achieves a $\sim 0.2\%$ higher accuracy. When ciphertext-wise pruning is applied, the runtime speed-up is the most significant, achieving a $6.5\times$ speed-up over HETAL with only $0.2\%$ accuracy drop.

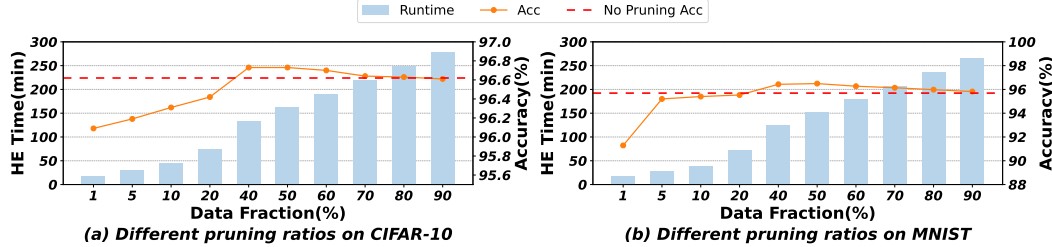

Figure 4: Private training time and accuracy with different fractions of data of (a) CIFAR-10 dataset and (b) MNIST dataset

**Ablation on the pruning ratio.** In Figure 4, we demonstrate the effectiveness of the proposed encrypted data pruning method under different data pruning ratios. In Figure 4 (a), we demonstrate the training time and accuracy achieved using different fractions of the CIFAR-10 dataset. Surprisingly, we find that even using only $1\%$ of the data, the accuracy drop is only around $0.6\%$, and the training can be sped up by $\sim 16\times$. Training with the $40\% \sim 70\%$ of the data even leads to higher accuracy than training with the full dataset. The same phenomenon is observed in plaintext data pruning [9]. This is because some noisy or low-quality samples are excluded from training while retaining enough informative samples are maintained. We note that we can achieve $\sim 2.2\times$ speed up when training with $40\%$ of the data without any loss of accuracy on the CIFAR-10 dataset. In Figure 4 (b), we experiment with the MNIST dataset and observe a similar trend. On the MNIST dataset, using only $1\%$ of the data achieves $91.29\%$ accuracy with $\sim 15\times$ speed-up. Yet, using $5\%$ of the data achieves a significantly higher accuracy of $95.2\%$, which is only $0.5\%$ lower than training with the full dataset.

Table 3: Privately Training an MLP from scratch under different data pruning ratios.

| Method | | 1% | 5% | 10% | 20% | 40% | 50% | 60% | 70% | 80% | 90% |
|---|---|---|---|---|---|---|---|---|---|---|---|
| Acc. | Acc(%) | 93.23 | 97.12 | 97.39 | 98.38 | 98.52 | 98.55 | 98.5 | 98.48 | 98.45 | 98.45 |
| | $\Delta Acc.$ | -5.26 | -1.37 | -1.1 | -0.11 | +0.03 | +0.06 | +0.01 | -0.01 | -0.04 | -0.04 |
| Runtime(h) | Time(h) | 32.25 | 110.61 | 208.56 | 404.46 | 796.26 | 992.16 | 1188.06 | 1383.94 | 1579.88 | 1775.72 |
| | speed up | 60.8$\times$ | 17.2$\times$ | 9.4$\times$ | 4.8$\times$ | 2.5$\times$ | 1.9$\times$ | 1.7$\times$ | 1.4$\times$ | 1.2$\times$ | 1.1$\times$ |

**Training from scratch.** We show the performance of encrypted data pruning in the training from scratch setting in Table 3. We train a 3-layer MLP on the MNIST dataset. We set the pruning

frequency as 10 epoch. When training with only $1\%$ data, the end-to-end training time could be $60.8\times$ faster with an accuracy drop of $5.26\%$. Increasing the training data fraction can effectively improve the test accuracy. When using $40\%$ and $50\%$ fraction of data, the training accuracy is $0.03\% \sim 0.06\%$ higher than training with the full dataset. We remark that warm-start strategy can also improve the test accuracy.

## 5  Discussion

**Broader Impact.** In this paper, we propose a framework that enables encrypted data pruning during confidential training. The proposed techniques can accelerate the private training without sacrificing the model accuracy. By incorporating data homomorphic friendly data pruning techniques, our framework makes the HE-enabled private training more practical while ensuring the data privacy.

**Limitations.** (1) More Model Support. Currently, the proposed methods have only been applied to simple models like MLP. Extending the encrypted data pruning to private training on larger models like CNNs and Transformers can enhance its utility. (2) More Dataset Support. In this paper, we have experimented on relatively small datasets. Extending to larger datasets will enhance its utility.

## 6  Conclusion

The paper presents a novel framework for encrypted data pruning aimed at enhancing private training of deep neural networks. Plaintext data pruning methods offer limited benefits in encrypted settings due to their lack of optimization for cryptographic processes. Our approach introduces crypto-oriented optimizations, including HE-friendly score, client-aided masking, and ciphertext-wise pruning, which effectively harness the potential of data pruning, achieving up to a 16-fold acceleration in training times without compromising accuracy.

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

# Appendix

## A  Encrypted data pruning protocol

In this section, we detail the proposed encrypted data pruning protocol. The main overhead associated with performing performing encrypted data pruning is determined by Step 3.(d) in Figure 5.

---

**Parties:** Client $C$, Server $S$.
**Input:** $C$ holds a private dataset $D$, with $N$ samples and labels $\{(X_i, Y_i)\}_{i=0}^{N-1}$.
**Output:** $C$ learns a trained model $\mathcal{M}$ and $S$ learns nothing.

---

**Protocol:**

1: $C$ encrypts the dataset with HE.Enc, and sends the encrypted dataset $\bar{D} = \{\bar{X}_i, \bar{Y}_i\}_{i=0}^{C-1}$ to the server.

2: $S$ initializes $\mathcal{M}$ with random parameters $W_0$ and chooses the hyperparamerters for training: the learning rate $\eta$, pruning ratio $p$ and the pruning frequency $\Delta\tau$.

3: For each epoch $\tau$, repeat the following:

    (a) $S$ performs forward pass homomorphically: $P = \mathsf{HE.SoftMax}(\mathcal{M}(X, W))$

    (b) $S$ computes gradients and updates model:
$$W_{\tau+1} \leftarrow W_\tau + \eta\nabla_W, \text{ where } \nabla_W \mathcal{L}_{CE} = \frac{1}{N}(P - Y)^T X$$

    (c) $S$ sends $P$ to $C$, $C$ determines early stop.

    (d) If $(\tau \bmod \Delta\tau == 0)$, $S$ performs data pruning as follows:

        (i) $S$ computes the importance score: $sc\bar{o}re = \mathsf{Prune.Eval}(\bar{P}, \bar{Y})$

        (ii) $S$ sends $sc\bar{o}re$ to $C$, can $C$ computes a pruning mask $mask$.

        (iii) $S$ removes unimportant samples: $\bar{D}' = \mathsf{Prune.Remove}(\bar{D}, mask)$

        (iv) Continue step 3(a) to train the model on the pruned dataset $\bar{D}' = \{\bar{X}'_i, \bar{Y}'_i\}_{i=0}^{C'-1}$

4: $S$ sends the trained model $\mathcal{M}$ with encrypted optimal weights $\bar{W}^*$ to $C$. $C$ decrypts the model with HE.Dec and obtain the final model $\mathcal{M}$ with optimal plaintext weights $W^*$.

Figure 5: Private Data Pruning Protocol.

## B  Analysis of Client-aided Masking

In this section, we offer more details about the proposed Client-aided Masking (CAM).

The main privacy issue is that (1) the server may learn private information based on the meta information, (2) the server may crack the secret key. Exposing the pruning mask $mask$ to the server only reveals the location of unimportant data samples, while all samples are still encrypted by HE, hence the server still learns nothing about the data sample itself, which avoids the security issue (1). One of the most recent and well-known attacks is the IND-CPA+ [43]. However, it operates under the assumption that the server can ask the client to decrypt specific ciphertexts. The assumption is not upheld in Algorithm 2. Since the server can not specify the ciphertext to be decrypted and that the client only sends the server the resulting mask, rather than the decrypted scores.

The communication depends on the level of ciphertexts. As shown in Table 4, the lower level the ciphertext is at, the smaller the ciphertext is. Therefore, the server can set the ciphertext to $L = 0$ to reduce the communication overhead.

Table 4: Ciphertext sizes under different levels.

| $N = 2^{16}$ | $L = 0$ | $L = 1$ | $L = 2$ | $L = 3$ | $L = 4$ | $L = 5$ |
|---|---|---|---|---|---|---|
| Ciphertext Size(MB) | 1.01 | 2.03 | 3.02 | 4 | 5.02 | 6 |

**Algorithm 4:** Ciphertext-wise Pruning

---

**Input**  :The encrypted dataset $\bar{D} = \{\bar{X}_i, \bar{Y}_i\}_{i=0}^{C-1}$ and pruning masks $mask = \{M_i\}_{i=0}^{C-1}$, where
$\quad\quad\quad M_i \in \{0,1\}^B$ and $B$ is the number of samples in each ciphertexts.
**Output** :The pruned encrypted dataset $\bar{D}' = \{\bar{X}'_i, \bar{Y}'_i\}_{i=0}^{C'-1}$, where $C' \le C$.
$\bar{D}' \leftarrow \emptyset$;
**for** $i \leftarrow 0$ **to** $C - 1$ **do**
$\quad\lfloor\ spasity_i \leftarrow B - \sum_{j=0}^{B-1} M_i[j]$;             `// compute the ciphertext-wise sparsity`
$ct\_queue \leftarrow Sort(spasity, \bar{D}, mask)$;
**while** $ct\_queue.is\_not\_empty()$ **do**
$\quad\vert\ ct\_front \leftarrow ct\_queue.pop\_front()$; `// The Trim() removes full and empty ciphertexts`
$\quad\vert\ $ `from the` $ct\_queue$
$\quad\vert\ $ **if** $ct\_front.sparsity == 0$ **then**
$\quad\vert\quad\vert\ \bar{D}' \leftarrow \bar{D}' \cup \{ct\_front.ct\}$;
$\quad\vert\quad\vert\ $ **continue**;
$\quad\vert\ $ **else**
$\quad\vert\quad\vert\ ct\_queue.push\_front(ct\_front)$;
$\quad\vert\quad\vert\ $ **break**;

**while** $ct\_queue.is\_not\_empty()$ **do**
$\quad\vert\ ct\_back \leftarrow ct\_queue.pop\_back()$;
$\quad\vert\ $ **if** $ct\_back.sparsity == B$ **then**
$\quad\vert\quad\vert\ $ **continue**;
$\quad\vert\ $ **else**
$\quad\vert\quad\vert\ ct\_queue.push\_back(ct\_back)$;
$\quad\vert\quad\vert\ $ **break**;

**while** $ct\_queue.is\_not\_empty()$ **do**
$\quad\vert\ ct_{front} \leftarrow ct\_queue.pop\_front()$;
$\quad\vert\ ct_{back} \leftarrow ct\_queue.pop\_back()$;
$\quad\vert\ slot_{empty} \leftarrow 0$;                `// The FZero() returns index of the first 0 in` $ct_{front}$
$\quad\vert\ $ **for** $slot \leftarrow 0$ **to** $B - 1$ **do**
$\quad\vert\quad\vert\ $ **if** $ct\_front.mask[slot] == 0$ **then**
$\quad\vert\quad\vert\quad\vert\ slot_{empty} \leftarrow slot$;
$\quad\vert\quad\vert\quad\vert\ ct\_front.mask[slot] \leftarrow 1$;
$\quad\vert\quad\vert\quad\vert\ ct\_front.sparsity\ -= 1$;

$\quad\vert\ slot_{used} \leftarrow 0$;                 `// The FOne() returns index of the first 1 in` $ct_{back}$
$\quad\vert\ $ **for** $slot \leftarrow 0$ **to** $B - 1$ **do**
$\quad\vert\quad\vert\ $ **if** $ct\_back.mask[slot] == 1$ **then**
$\quad\vert\quad\vert\quad\vert\ slot_{used} \leftarrow slot$;
$\quad\vert\quad\vert\quad\vert\ ct\_back.mask[slot] \leftarrow 0$;
$\quad\vert\quad\vert\quad\vert\ ct\_back.sparsity\ += 1$;

$\quad\vert\ slot_{used} \leftarrow \mathsf{FOne}(ct_{back})$;
$\quad\vert\ k \leftarrow slot_{used} - slot_{empty}$;
$\quad\vert\ M_t \leftarrow \{0\}^B$;                 `// The Mask() keep only the sample at` $slot_{used}$ `in` $ct_{back}$
$\quad\vert\ M_t[slot_{used}] \leftarrow 1$;
$\quad\vert\ ct_{align} \leftarrow ct_{back} \boxtimes M_t$;
$\quad\vert\ ct_{front} \leftarrow ct_{front} \boxplus \mathsf{HE.Rot}(ct_{align}, k)$;
$\quad\vert\ $ **if** $ct\_queue.is\_empty() \vee ct_{front}.sparsity == 0$ **then**
$\quad\vert\quad\vert\ \bar{D}' \leftarrow \bar{D}' \cup \{ct_{front}.ct\}$;
$\quad\vert\ $ **else**
$\quad\vert\quad\vert\ ct\_queue.push\_front(ct_{front})$;
$\quad\vert\ ct\_queue.push\_back(ct_{back})$;

**return** $\bar{D}'$

---

## C  Datasets and Hyperparameters

We detail how we divide the training set, validation set and test set in Table 5. For all tasks, we use a batch size of 128. For the transfer learning setting, we use a learning rate of $0.5$ and set the pruning frequency as every $\Delta_\tau = 5$ epochs. For the training from scratch setting, we use a leaning rate of

0.1 and set the pruning frequency as every $\Delta_\tau = 10$ epochs. Prior works [9, 12] adopt a warm start strategy, which trains the model for $10 \sim 20$ epochs on the full dataset. The warm start period is computation intensive as it uses the whole dataset. Therefore, we use a random start strategy and randomly choose a fraction of data samples according to the pruning ratio $p$.

Table 5: Dataset Distribution for Various Machine Learning Challenges

| Dataset | Total | Train | Validation | Test |
|---|---|---|---|---|
| MNIST | 70000 | 52500 | 7500 | 10000 |
| CIFAR-10 | 60000 | 43750 | 6250 | 10000 |
| Face Mask Detection | 4072 | 2849 | 408 | 815 |
| DermaMNIST | 10015 | 7007 | 1003 | 2005 |
| SNIPS | 14484 | 13084 | 700 | 700 |

## D  Ciphertext-wise pruning

In Algorithm 4, we detail the Trim(), FZero(), Fone() and Mask() functions, which are used during ciphertext-wise pruning. When $k > 0$, the server perform left rotation on $ct_{back}$ by $|k|$ sample slot. If $K < 0$, right rotation is performed.

