# OpenReview forum: "HEPrune: Fast Private Training of Deep Neural Networks With Encrypted Data Pruning"
_NeurIPS.cc/2024/Conference — NeurIPS 2024 poster_

### Official Review · Reviewer_ADwT · 2024-06-14

**Soundness:** 2
**Presentation:** 2
**Contribution:** 2
**Rating:** 6
**Confidence:** 4

**Summary:**

This paper proposes a data pruning algorithm for the training of Homomorphic Encryption (HE)-based neural networks. The authors introduce an HE-friendly importance score and client-aided masking to prune samples in the dataset. The authors further propose ciphertext-wise pruning to merge ciphertexts with empty slots, thereby reducing computational costs during training. Finally, the paper presents empirical studies to validate the effectiveness of the proposed data pruning method.

**Strengths:**

The main advantages can be listed as follows:

1.	The paper provides a new data pruning method for data encrypted by HE scheme. The authors propose the HEL2N score, which substitutes the $\ell_2$-norm in the EL2N score with $\ell_1$-norm, and the client will select important samples based on the score computed by server.
2.	The paper proposes the ciphertext-wise pruning, enabling the server to merge ciphertexts with empty slots with the communication of the client.
3.	The paper conducted experiments on five datasets and compares the proposed method with the HETAL method to demonstrate its effectiveness.

**Weaknesses:**

Despite many strengths, there are some issues with the paper as follows:

1.	The submission requires further revisions for clarity and consistency. At line 43 “methds” should be “methods”; At line 74 and Figure 1 “CIFAR10” should be “CIFAR-10” as in Section 4; At line 326 “Table2” should be “Table 2”; At line 340 “Figure 4(4)” should be “Figure 4(a)”; Figure 1 and 4 should have sub-captions denoting which subgraph is (a) or (b).
2.	The computation costs associated with data pruning raise concerns. As given by Eqn. (1), computing the HEL2N score involves multiple gradient computations for each sample, which is computationally intensive. Moreover, ciphertext-wise pruning seems to require a large number of rotations, which is also a very slow HE operation. If the data pruning process is time-consuming, it may negate the benefits, making it more efficient to train directly without pruning.
3.	The novelty of the paper is questionable. The HEL2N score primarily modifies the $\ell_2$-norm in the EL2N score to an $\ell_1$-norm, and directly computing the square of EL2N score seems to be faster than HEL2N score. The trick of masking is also common place in HE literature. Moreover, one advantage of HE-based method is that they do not require any communications between server and client. The requirement for client-server communication in client-aided masking and ciphertext-wise pruning could diminish the significance of the proposed method.

**Questions:**

1.	Why not directly compute the square of the EL2N score, which would avoid the need for computing the square root or the maximum value and is a simpler process?
2.	Is the running time for HE-based data pruning included in the total running time reported in the Tables in Section 4.2? If so, what proportion of the total running time does the HE-based data pruning constitute?
3.	How does the proposed method compare to a baseline that directly randomly samples a subset of ciphertext at each epoch, instead of computing importance score and merging ciphertext? I think this baseline is simpler and more efficient.

**Limitations:**

The authors addressed the limitations.

---

> ### Author Rebuttal · Authors · 2024-08-07
>
> We thank Reviewer ADwT for his/her thorough reading of the manuscript and constructive comments.
>
> Q1 Clarity and consistency
>
> We thank the reviewer for the thorough reading. We will fix the typos in the future version.
>
> Q2 The overhead of the proposed methods
>
> The results in Section 4.2 include the running time for HE-based data pruning. The proposed data pruning mainly contains (1) computing the HEFS score and (2) performing ciphertext-wise pruning. We show the proportion of the proposed methods of total runtime in Table 2(rebuttal pdf).
>
> We first benchmark the runtime of basic HE operations in our environment: addition:8ms, ciphertext-plaintext multiplication (cpMult):17ms, rotation:116ms, ciphertext-ciphertext multiplication (ccMult):146ms, max:1.37s and bootstrapping:35.89s.
>
> **HEFS.** The proposed HEFS does not rely on the gradients. Instead, HEFS can be computed via the error vector as shown in the Equation (2). To compute HEFS, we only need subtraction, max and the rotation operation. Computing HEFS on a single ciphertext takes 1.72~1.84s.
>
> **Ciphertext-wise Pruning.**
> While rotation is not fast compared with addition and cpMult, it is still faster than the ccMult in the CKKS scheme. Moreover, the rotation barely consumes the noise budget [1]. We remark that the non-linear SoftMax function is computed via approximation, which also relies on ccMult. The intensive evaluation of the ccMult in the forward pass and back propagation leads to frequent bootstrapping, which is the most expensive operation in practice.
>
> In the worst case, the ciphertext-wise pruning needs $O(p(1-p)N)$ rotations, where $N$ is the number of total data samples and $p$ the pruning ratio. Take the CIFAR-10 dataset for an example. We choose a batch size of 128, and the 43750 training data samples are encrypted in 342 ciphertext blocks. When the pruning ratio is 99\%, the ciphertext-wise pruning needs at most 430 rotations to combine the 342 sparse ciphertext blocks into 4 dense ones, which takes 0.04h. In contrast, evaluating one epoch without pruning takes more than 20h.
>
> We remark that the main bottleneck of private training remains to be the evaluation of complex non-linear functions like the SoftMax function and the bootstrapping. We believe that optimizing these operations could further accelerate private training. This is orthogonal to the optimizations discussed in this paper.
>
>
> Q3 The effectiveness of the proposed HEFS.
>
> In Figure 2(rebuttal pdf), we compare HEFS with (1) the square of the EL2N score; (2) randomly dropping ciphertexts. However, these two methods yield inferior efficiency / utility trade off compared with the proposed HEFS.
>
> **The square of EL2N.**
> The HEFS generally has a better accuracy than the squared EL2N. HEFS outperforms the squared EL2N by a noticeable margin when the pruning ratio is high, i.e., the remaining data is below 0.2. This is because the squared EL2N score enlarges the importance of samples that are predicted wrong. These samples can be noisy samples [2]. Emphasizing these samples too much makes the selected data less informative, which leads to an inferior accuracy. As the pruning ratio goes down, the proportion of such noisy samples decreases. Accordingly, the performance of the squared EL2N becomes closer to HEFS. In Table 1(rebuttal pdf), we show that computing the score is not the bottleneck and the savings from using the squared EL2N is marginal.
>
> **Random pruning.** While random pruning strategy is simple, it ignores the importance of data samples. This leads to significant deterioration of accuracy, especially when the remaining fraction is low.
>
> Q4 The novelty and the significance.
>
> To the best of our knowledge, we are the first to explore encrypted data pruning to accelerate private training. Performing data pruning in the encrypted state is non-trivial. Directly adapting plaintext pruning methods is not viable. To start with, the importance score in the plaintext often involves complex nonlinear functions that are expensive in HE. Moreover, sorting samples are considered free in plaintext, but sorting in the encrypted state is expensive. Most importantly, the sample-wise pruning in plaintext leads to sparse ciphertexts and fails to effectively accelerate private training. We propose a series of HE-aware optimizations including HEFS, client aided masking and ciphertext-wise pruning. Our methods boost the efficiency of private training significantly without sacrificing accuracy.
>
> Moreover, we would like to remark that EL2N has been treated as a static data pruning method. Prior works have assumed either that it remains unclear if EL2N reduces the total training time [3] or that EL2N cannot be used to accelerate training [4]. In this work, we innovatively adapt it to the dynamic data pruning setting and show its effectiveness.
>
> Q5 The non-interactivity of HE
>
> Private training with the proposed encrypted data pruning remains largely non-interactive compared with the MPC-based methods. This is because all HE computation is performed solely by the server. The client is only required to be online for less than 0.1% of the total training time, to handle the early stop signal and pruning mask. As shown in prior works like HETAL, these communications are both secure and necessary.
>
> To sum up, we show that the squared EL2N and randomly pruning are less effective than the proposed HEFS. Additionally, the proposed HEFS and ciphertext-wise pruning are efficient at runtime. We sincerely hope our responses have addressed the reviewer's concerns.
>
> Reference
>
> [1] Bossuat, Jean-Philippe, et al. Efficient bootstrapping for approximate homomorphic encryption with non-sparse keys.
>
> [2] Paul, Mansheej, et al. Deep learning on a data diet: Finding important examples early in training.
>
> [3] Truong, Thao Nguyen, et al. KAKURENBO: adaptively hiding samples in deep neural network training.
>
> [4] Qin, Ziheng, et al. Infobatch: Lossless training speed up by unbiased dynamic data pruning.

---

> > ### Comment · Reviewer_ADwT · 2024-08-10
> >
> > Thank you for the detailed response. Considering the authors' new clarification and overall contribution of the paper, I am willing to improve my score.

---

> ### Author Response · Authors · 2024-08-10
>
> We sincerely appreciate the reviewer's thoughtful feedback and are grateful for the increased score. Thank you for your consideration and support.

---

### Official Review · Reviewer_LFsM · 2024-06-30

**Soundness:** 3
**Presentation:** 3
**Contribution:** 3
**Rating:** 7
**Confidence:** 3

**Summary:**

This paper focuses on the scenario where the client encrypts the model and dataset with homomorphic encryption and outsources them to the server for training. It accelerates the training process through dynamic data pruning. This paper makes the following three contributions:
First, this paper is the first to use dynamic data pruning to accelerate model training in homomorphic encryption (HE) scenarios. Second, because using the plaintext data pruning method in the HE scenario incurs significant overhead, this paper proposes an HE-friendly method for evaluating the importance of data samples. Lastly, because of the high cost of sorting in HE, this paper proposes that the client undertake this part of the computation. Additionally, since a single SIMD ciphertext can contain multiple data samples, pruning may not reduce the number of ciphertexts, even though the samples within each ciphertext become more sparse. To address this issue, the paper proposes to combine several sparse ciphertexts to reduce HE computation.

**Strengths:**

1) This paper is the first to apply dynamic data pruning to accelerate model training in HE scenarios.
2) It introduces a HE-friendly important score to make data pruning more efficient.
3) This paper uses ciphertext-wise pruning to reduce the number of ciphertexts while keeping detailed information.

**Weaknesses:**

1) The HE-friendly score needs more explanation. In this paper, the score is directly introduced without any theoretic proof of its effectiveness.
2) The work is a bit incremental. Applying data pruning in the HE scenario doesn't seem very challenging, and there is no significant difference between data pruning in plaintext and ciphertext.

**Questions:**

1) Is there any theoretic proof of the effectiveness of the proposed HE-friendly score?

---

> ### Author Rebuttal · Authors · 2024-08-07
>
> We thank Reviewer ADwT for his/her thorough reading of the manuscript and constructive comments.
>
> Q1 The effectiveness of HEFS
>
> The proposed HE-friendly score relies on the observation that the importance of a data sample can be quantified by its gradients [1]. We denote the input vector as $x\in\mathbb{R}^d$ where $d$ is the input dimension and the one-hot label as $y\in \{0,1\}^K$ where $K$ is the number of class. We denote the logit outputs of the model as $f(x;w)$ and the prediction vector as $p(x;w)=\sigma(f(x;w))$, where $\sigma(\cdot)$ is the SoftMax function. We denote the cross-entropy loss as $\ell(p,y)=\sum_{i=0}^{K-1}y^{(i)}\log p^{(i)}$.
>
> We denote a minibatch of $B$ samples as $S=\{(x_i,y_i)\}_{i=0}^{B-1}$.
>
> We denote a single sample's gradient to the weights at time $t$ as $g_t(x,y)=\nabla_{w_t}\ell(p,y)$. The change of a sample's contribution to training at time $t$ can be quantified by the time derivative of the loss on the sample, i.e., $\Delta_t((x,y), S)= -\frac{d \ell(p,y)}{dt}$. Accordingly, we can evaluate the importance of some data sample $(x_k,y_k) \in S$ by investigating how removing the $(x_k,y_k)$ from the minibatch change the loss of other samples. The impact of removing the sample $(x_k,y_k)$ is bounded by its gradients. More formally, for $\forall (x_i,y_i)\in S$ where $i\neq k$, we have:
> $\| \Delta_t((x_i, y_i), S) - \Delta_t((x_i, y_i), S_{\neg k}) \|_2 \leq c \| g_t(x_k, y_k) \|_2 \cdots (1)$
>
> We provide a proof of the correctness of the Equation (1) as follows. By the chain rule, we have $\Delta_t((x_i, y_i), S) =  -\frac{d \ell(p,y_i)}{dt} = -\frac{d \ell(p,y_i)}{dw_t}\frac{dw_t}{dt}$. When the weights are updated via SGD, we have $\frac{dw_t}{dt}=-\eta\sum_{(x_j,y_j)\in S}g_t(x_j,y_j)$ where $\eta$ is the learning rate. Accordingly, the change of the loss of the sample $(x_i,y_i)$ is $-\frac{d \ell(p,y_i)}{dw_t}\cdot\eta\sum_{(x_j,y_j)\in S}g_t(x_j,y_j) + \frac{d \ell(p,y_i)}{dw_t}\cdot\eta\sum_{(x_j,y_j)\in S_{\neg k}}g_t(x_j,y_j) = \eta\frac{d \ell(p,y_i)}{dw_t}g_t(x_k, y_k)$. By the submultiplicative property of norms, we have $\| \Delta_t((x_i, y_i), S) - \Delta_t((x_i, y_i), S_{\neg k}) \|_2 \leq \eta \| \frac{d \ell(p,y_i)}{dw_t} \|_2 \| g_t(x_k, y_k) \|_2$. Since $\eta$ and $\frac{d \ell(p,y_i)}{dw_t}$ is independent of $k$, we can set $\eta \| \frac{d \ell(p,y_i)}{dw_t} \|_2$ as a constant $c$. Thus, we have proved the correctness of the Equation (1).
>
> The above bound can be simplified. Specifically, by the chain rule, we have $g_t(x,y)=\frac{d \ell(p,y)}{d f}\frac{d f}{d w_t}$. Given the number of class $K$, the bound can be written in the logit-wise form as $\sum_{i=0}^{K-1}\| \nabla_{f^{(i)}}\ell(p,y)^T\nabla_{w_t}f^{(i)}(x;w_t) \|_2$.
>
> Under the cross entropy loss, the derivative with respect to the $i$-th logit is $p(x;w_t)^{(i)}-y$. It was observed that logit gradients $\{\nabla_{w_t}f^{(i)}(x;w_t)\}_{i=0}^{K-1}$ are generally orthogonal among classes [2,3]. Thus, the bound can be simplified as $\| p(x;w_t)-y \|_2$, which is the EL2N score.
>
> The proposed HEFS further simplifies the computation as the $\ell_1$-norm of the error vector. We observe that the $\ell_1$-norm of the error vector has identical properties as the $\ell_2$-norm. If a sample's prediction vector is largely similar to the label vector, it will have both a small EL2N score and a HEFS score. This indicates that the sample is easier to learn and less informative for the training process. Our experiments also supports our analysis. HEFS can effectively select informative data samples to accelerate training without compromising the accuracy. Additionally, while computing the square root in EL2N is expensive in HE, HEFS can be computed in HE much more efficiently.
>
> Q2 Why encrypted data pruning is challenging
>
> While the idea of leveraging data pruning to accelerate the private training seems straightforward, we note that encrypted data pruning differs from data pruning in the plaintext in several key aspects.
>
> Computing these metrics such as the entropy loss and the $\ell_2$-norm is considered as free in the plaintext. However, evaluating the complex non-linear functions in HE is non-trivial. Evaluating a single square root function on a ciphertext can take up to 2 minutes. This implies that simple metrics like EL2N could negate the benefits of data pruning. We highlight that HEFS rely solely on the prediction vector and the label vector, which are directly available from the private training process. Evaluating the HEFS score only requires subtraction, max and the rotation operation, which takes only 1.72~1.84s for one ciphertext.
>
> Another key difference is that removing the less important data samples directly reduce the computation overhead of training in the plaintext. However, merely excluding the less important data samples does not effectively reduce the overhead of private training. This is because the runtime of private training is determined by the number of ciphertexts, and one ciphertext can contain multiple data samples. Performing sample-wise data pruning leads to a large number of sparse ciphertext. We introduce ciphertext-wise data pruning to effectively reduce the number of ciphertext and boost the efficiency of private training.
>
> The proposed data pruning methods are lightweight, taking only as little as 0.4%~2.5% total runtime, while effectively accelerating private training. We believe this work enables more practical private training in real-world. We thank the reviewer's suggestions and will incorporate the detailed analysis of HEFS in the future version for better clarity.
>
> Reference
>
> [1] Paul, Mansheej, et al. Deep learning on a data diet: Finding important examples early in training.
>
> [2] Fort, Stanislav, et al. Deep learning versus kernel learning: an empirical study of loss landscape geometry and the time evolution of the neural tangent kernel.
>
> [3] Fort, Stanislav, et al. Emergent properties of the local geometry of neural loss landscapes.

---

### Official Review · Reviewer_FEVE · 2024-07-14

**Soundness:** 3
**Presentation:** 3
**Contribution:** 3
**Rating:** 5
**Confidence:** 4

**Summary:**

1. The paper introduces a Homomorphic Encryption (HE)-based confidential training framework that enhances training efficiency through encrypted data pruning.
2. The paper proposes HE-Friendly Score (HEFS), an enhancement over the existing EL2N score, to efficiently assess the importance of encrypted data samples.
3. Due to the high complexity of sorting scores and calculating the pruning mask on the server, the paper introduces a method that generates data pruning masks with the assistance of the client, enabling the server to perform pruning.
4. The paper proposes a method for pruning at the ciphertext level to reduce sparsity in the encrypted data, thereby accelerating the training process.
5. The performance of HEFS, CAM, and CWP is evaluated on diverse datasets such as MNIST, CIFAR-10, Face Mask Detection, DermaMNIST, and SNIPS. The results are compared with the previous method, HETAL (ICML2023), to demonstrate improvements in training speed and accuracy.
6. The experimental results indicate that the proposed methods can accelerate confidential training by up to 16 times with minimal loss in accuracy.

**Strengths:**

1. This paper tackles the novel problem of accelerating confidential training through encrypted data pruning, a topic that appears to have not been previously explored in existing research.
2. The methodologies and experimental procedures are clearly explained, ensuring the reproducibility of the results by providing their code (although I have not tested the code yet).
3. Considering that HE training is a very challenging subject due to the computational complexity of operations on homomorphically encrypted data, it is noteworthy that the authors have implemented detailed techniques such as pruning within the homomorphic encryption framework for the first time. However, it is crucial that the design carefully considers both security and performance limitations.

**Weaknesses:**

1. There are concerns regarding the privacy threat setting in this paper. The focus is solely on the importance of the client's data privacy, without explicitly considering the server's model privacy. In other words, the server's model is assumed to be publicly available information. This assumption is reflected in the final step, where the client recovers the weights of the trained model and sends them back to the server.
2. If the server's model is publicly available, it would be more efficient for the client to process the data in plaintext after receiving the pretrained model. If the scenario involves a massive pretrained model, such as a large language model (LLM), which individual clients cannot train, then training such an LLM in an encrypted state would require at least 1000 times more computation on the server due to the difference in computational overhead between homomorphic encryption and plaintext operations. This level of computation may be unmanageable, and the final decryption of the model by the client would also be infeasible due to the enormous size (100 trillion parameters) of the model.
3. If the primary concern is the client's privacy, it would be significantly more efficient to have the client train on a pretrained model in plaintext, use federated learning, or adopt other methods rather than struggling with encrypted training on the server, as proposed in this paper. The bottom line is that if the server's model privacy is not considered in the security threat model, it is questionable whether this approach is practical or appropriate.
4. The HE.cmp operation does not yield a precise 1 or 0; rather, it produces a fractional value when the two compared values are similar. During the sorting process in Algorithm 1, swapping based on HE.cmp might introduce noise if the score differences are not substantial. This could affect performance. To avoid this, HE.cmp would need a high-degree approximation. The paper does not provide sufficient information on HE.cmp, and additional explanation would be beneficial.
5. Due to the overhead of homomorphic encryption operations, involving the client in the training process because of the complexity of sorting diminishes the method's utility. While the paper does not consider the server's model privacy, allowing the client to access intermediate values during training does not pose an additional security risk. However, in scenarios where the server's model privacy is a concern, this method could enable the client to gain critical information about the model. Increasing the client's role in plaintext processing during training could significantly reduce the server's burden and enhance overall performance. In extreme cases, the client could potentially handle the entire training process in plaintext. The paper needs to explain why the client should only assist with sorting.

**Questions:**

Is it true that the pruning mask is also encrypted? The distinction between encrypted and non-encrypted elements in the algorithm is not entirely clear. While an overline appears to indicate encrypted values, there are instances where this is not consistently applied.
In scenarios where the server does not send scores to the client, the mask would presumably remain encrypted. If this is the case, how does the server determine which parts to prune, and how does it use rotation to merge ciphertexts and create pruned ciphertexts? Is it assumed that the client must decrypt the data and create the mask in plaintext? If so, the paper should explicitly state this assumption.

**Limitations:**

They addressed it adequately.

---

> ### Author Rebuttal · Authors · 2024-08-07
>
> Q1 Clarification on the threat model
>
> We would like to clarify some misunderstandings about the privacy threats this paper focuses on. The proposed method protects both the data privacy and the model privacy. In our threat model, both training data and the model weights belong to the client. The client does not have adequate resource or knowledge for training and outsources the training task to the server. Neither the training data nor the model weights are revealed to the server at any time. All the intermediate features and gradients are also encrypted and protected by HE. This threat model is commonly used in private training works [1].
>
> Weakness 1. Since the model belongs to the client, it is reasonable that the client can decrypt the model. Additionally, we note that the client does not need to send the decrypted model to the server. Since the model is already well-trained, the client can keep it for local use. Alternatively, if the client requires the server to provide private inference service on the trained model, it is possible to store the encrypted final model on the server side.
>
> Weakness 2.  As mentioned above, the model does not belong to the server and is not publicly available.
>
> Weakness 3. As mentioned above, the goal and privacy threats are different from those in federal leaning. In our threat model, the client outsources the training task to the server because the client does not have enough resource or knowledge for training. In the meantime, the gradients are encrypted and not revealed to the server. Federal learning typically needs local training on the client side and does not directly protect the gradients.
>
> Q2 The security of the pruning mask
>
> (1) Unencrypted mask. In response to weakness 5, we explain why client aided masking is secure and reasonable. As explained above, the both the data and the model belong to the client. Therefore, revealing the mask to the client does not pose security risk against the data privacy or model privacy. In our client aided masking algorithm, the importance score is revealed to the client. The client performs a quick selection algorithm to find the score threshold and generate a pruning mask. This happens in the unencrypted state on the client side.
>
> During the client aided masking and ciphertext-wise pruning, the plaintext pruning mask is revealed to the server to enhance the efficiency of data pruning on the server side. The server does learn the sparsity information of the training dataset implied by the pruning mask. However, since all the training data and model weights are strictly encrypted on the server side, the privacy of client's data and model is still protected. Similar to prior works [1], we allow revealing necessary meta information about training like the the size of the dataset or ciphertexts, the early stop signal and the pruning mask. Sharing these information does not pose direct security risk against the client’s data privacy or model privacy.
>
> (2) Encrypted mask. We explain how the server can remove unimportant samples using encrypted mask without client aided masking. Performing data pruning without the client's aid and via an encrypted mask is possible. The server can first compute the importance score in the encrypted state and generate the encrypted pruning mask as shown in Algorithm 1. With the encrypted pruning mask, the serve can make the slots of unimportant samples empty via homomorphic multiplication and generate sparse ciphertexts. However, since the mask is encrypted, the server cannot learn which samples should be pruned. The server cannot merge the sparse ciphertexts via rotation either. This is because the sparsity of each ciphertext remains unknown to the server, and the server can not determine how to rotate each ciphertext to create pruned ciphertexts. Therefore, while data pruning with encrypted mask is possible, it incurs prohibitive overhead during homomorphic sorting in practice and is barely useful.
>
> Our experiments are indeed based on the client aided masking and the unencrypted pruning mask. We will state this more explicitly for better clarity.
>
> Q3 The approximated comparison in CKKS
>
> The comparison function in the CKKS scheme is approximated and not accurate. We approximate the sign function to compute the comparison function and the max function in equation (3). Both the max and the comparison function can be constructed via the sign function and have similar complexity [3]. Specifically, the max function can be defined as $max(a,b) \approx \frac{(a+b)-(a-b)sign(a-b)}{2}$, where $sign(x)$ returns 1 if $x>0$, 0 if $x==0$ and $-1$ if $x<0$. We use a composition of polynomials to approximate the sign function, i.e., $sign(x) \approx f(g(x))$, where $f(x)=f(x) = 8.83133072x - 46.45750399x^3 + 83.02822347x^5 - 44.99284778x^7$ and $g(x)= 3.94881885x - 12.91030110x^3 + 28.08653622x^5 - 35.59691490x^7 + 26.51593709x^9 - 11.41848894x^{11} + 2.62558444x^{13} - 0.24917230x^{15}$. The approximation error of the max function $e = \left| \max(a,b) - \frac{(a+b) - (a-b)\text{sign}(a-b)}{2} \right|$ is bounded by $2^{-8}$ [3]. While increasing the number of compositions and the degree of each polynomial can lead to an even smaller bound, we find that above approximation is precise enough for computing the HEFS score. Prior works [1] find that polynomials of even lower degree is enough for private training.
>
> We hope our responses address the reviewer's concerns. We will add the details about the max and comparison function to Appendix in future version for better clarity.
>
> [1] Lee, Seewoo, et al. HETAL: efficient privacy-preserving transfer learning with homomorphic encryption.
>
> [2] Lee, Eunsang, et al. Minimax approximation of sign function by composite polynomial for homomorphic comparison.

---

### Official Review · Reviewer_98ig · 2024-07-25

**Soundness:** 3
**Presentation:** 3
**Contribution:** 3
**Rating:** 5
**Confidence:** 4

**Summary:**

This paper presents a method for pruning data in a utility-preserving way under homomorphic encryption, evaluating the method to demonstrate that the savings from training on pruned data outweighs the costs of encrypted data pruning computations. The methods for determining how relevant data items are to improving training performance are similar in spirit to those in the active/few-shot learning literature, but the paper does not explicitly draw this parallel.

Although the paper offers a concrete threat model related to "private training" (where model training is outsourced to a third party), several aspects of the threat model seem not to achieve the stated goal of limiting the outsourcing party/vendor's ability to learn useful facts about the data. As the current state-of-the-art is to bind third-party infrastructure providers contractually, I'd like to see a map from how the approach in this paper, in a threat model that is weak, could be strengthened to a threat model that would obviate these purely contractual limitations. For example, there are standard constructions to move from honest-but-curious models to models where the third party is more adversarial (although some reliance must always be assumed in the case of outsourced computation). More difficult are the problems of separating data flows from inferences about the content of training data through various sorts of indirect leakage (training time, ciphertext size, dependence of the presence/absence of the allowed "early stop" signal, etc). Although this problem is difficult in general, I suspect there are ways to organize the threat model here so that it can make stronger claims around solving them. At a minimum the threat model should declare indirect data flow out of scope while recognizing that it can leak data items to the outsourcing partner.

A few structural aspects of the paper confuse an otherwise solid presentation: a core assumption in the setup of the model in which the protocol is used is that the outsourcing partner will compare the sorted data items (under encryption) to a threshold importance score determined by the utility loss of pruning, but how this threshold is set/computed is left open (an experiment uses an ex vacuo value of 0.9 for this parameter, but why is not explained even in this concrete context!); although it is clear that the goal is only to outsource training, it is not clear that an organization without the infrastructure capability for training will have infrastructure capability for things like serving - this should be declared out of scope or unpacked/discussed a bit; approximations made to simplify computation under encryption, such as the replacement of $\ell_2$ with $\ell_1$ norm at 204-205, are not directly evaluated or justified; in general, the grouping of samples into batch ciphertexts is assumed but not explained - the paper should explain its necessity and the benefits it provides vs. the simple solution of putting each sample in its own ciphertext.

Although the evaluation is valuable in supporting the core claims of the paper, there are some structural issues there as well: although the experiments characterize the tradeoff between utility and pruning ratio, this tradeoff is very different for the two example datasets. How general ought a tradeoff curve for this be? How data dependent might it be? Relatedly, might pruning affect performance for different classes differentially, especially in situations where class balance is poor? Much of the "fairness" literature focuses on ways that aggregate analysis breaks down when distinct classes might or ought to be treated differently by the model. Does this affect the analysis? Could it in some cases?

Last, I observe that 4MB of communication overhead for a tiny database (CIFAR-10, 43750 samples) is manageable but there is no discussion of scaling here. Are the proposed applications small like this? If not, at what point is the overhead too much? Does scale cause this to break down? I here recognize that the paper inherits the inherent inefficiency of FHE constructions.

**Strengths:**

* The research question is well motivated and the solution a useful tool to making private outsourced training a more realistic option. I am not well versed enough in the FHE training literature to evaluate the novelty of this specific approach, but the core idea is sound.
* Evaluations justify the theoretical claims nicely, even where improvements are available.
* The overall argumentation is strong, even if some details are never defined or explained as noted in the summary.

**Weaknesses:**

* The summary notes several places where definitions can be sharpened or details can be re-ordered to improve the presentation.
* There are a handful of places where some copyediting would improve the presentation, although the high-level argument structure is in general strong.
* The key question of how the pruning ratio is determined must be explained, since it is an input to most of the provided private training algorithms and also a key determinant of the model in which the protocol is meant to be used.

**Questions:**

* How does the pruning ratio get determined in practice? Is it necessary to do many private training runs and measure utility on the resultant models? How does this cost compare to the cost of non-private training?
* Is the scale problem being swept under the rug at 237-240 an issue for larger data sets, or is the idea that this should only apply to data sets at the scale demonstrated? How does communication overhead scale as the training runs become larger?
* Can the issue of indirect data leakage be managed somehow, or must it simply be assumed away in the security model? It's a hard problem, so solving it may well be out of scope here, but also past computation-on-encrypted-data methods have failed catastrophically due to indirect leakage problems so it's necessary to say something about it. What can or should the paper say?

**Limitations:**

As noted in the summary, there are places where limitations could be more clearly expressed, for example with regard to indirect data leakage, with regard to the determination of the critical pruning ratio parameter, and also with regard to aggregated analysis and generality of the pruning/utility tradeoff.

---

> ### Author Rebuttal · Authors · 2024-08-07
>
> We thank Reviewer 98ig for his/her careful reading of the manuscript and constructive comments.
>
> Q1 Deciding the pruning ratio.
>
> In practice, the pruning ratio should be determined by the client during the client aided masking process. As shown in Figure 1 (in rebuttal pdf), a moderate pruning ratio around 0.5 has close accuracy to training with the full dataset across different datasets. Therefore, like the plaintext data pruning works, one straightforward strategy can be choosing a fixed pruning ratio [1]. Recent works show that one can also choose a set of pruning ratios for different epochs [2]. Moreover, the server can store checkpoints after each data pruning. Since the client can observe the test accuracy, the client can also roll back to certain checkpoint if needed. One should avoid running private training multiple times. This is because private training is orders of magnitudes more expensive than the non-private training.
>
> Q2 The communication scalability.
>
> The proposed data pruning methods are scalable to large datasets. The communication complexity of the proposed method is linear with respect to the number of total data samples and the frequency of the pruning. For the ImageNet, performing one encrypted data pruning over 1,281,167 training samples would only incur around 40MB communication overhead. Note that data pruning is performed every $\Delta_{\tau}$ epochs, so the overall communication are few multiples of 40MB.
>
> The major challenge of scaling to larger datasets is the inefficiency of the HE constructions. As far as we know, the latest HE-based private training works still focus on small datasets like CIFAR-10 and MNIST [3]. Securely training on ImageNet can still take years even with GPU acceleration and more efficient cryptographic tools [4].
>
> Q3 Threat model and indirect data leakage.
>
> The threat model can be strengthened via additional integrity techniques like zero-knowledge proofs (ZKP). HE alone cannot ensure the integrity of the computation on the server side and we can introduce ZKP to verify the correct execution on the server side. Making HE verifiable via ZKP is an active research topic that is currently being explored [5]. In addition to ZKP, other integrity techniques such as Message Authentication Codes and Trusted Execution Environments can also be utilized to defend against malicious adversaries.
>
> Protecting the indirect information, such as the training time and the size of the dataset and ciphertext, is indeed out of scope for this work. In the meantime, we believe the security risk of such indirect information leakage is limited, since all data and model weights are strictly encrypted during the whole training phase. In this work, we focus on protecting client’s data privacy and model privacy in the semi-honest setting, like prior works on private training [3]. Such a setting is reasonable because as a service provider, the server is motivated to adhere to the protocols to guarantee the quality of service. We will make more explicit declarations for better clarity.
>
> Q4 Generality issue
>
> We show the pruning/utility tradeoff on more datasets in Figure 1 (rebuttal pdf). The accuracy does differ for different datasets. We notice that extremely high pruning ratios like 0.99 only work for CIFAR-10 dataset and do not generalize well. On the other hand, moderate pruning ratios demonstrate reasonable pruning/utility tradeoff across all datasets. When we keep more than 10% data, the accuracy is only 0.05%~0.31% lower than training with full dataset. When we keep around 50% data, the accuracy can be even higher.
>
> Q5 Explanations on details
>
> We thank the reviewer’s thorough reading and constructive advice on the copyediting.
>
> (1) The client's capability. The client does not have adequate resource or knowledge for training but has basic computation resource for at least encryption and decryption. The encryption and decryption algorithms are lightweight in HE, taking only 86ms and 49ms for one ciphertext respectively. Accordingly, it is reasonable to assume that the client can compute the pruning mask, which takes as little as 15ms for the CIFAR-10 dataset. Similarly, the client should be capable of computing the early stop signal [3].
>
> (2) The effectiveness of the HEFS score. A sample's importance can be quantified by the $\ell_2$-norm of its gradients, which can be approximated by the $\ell_2$-norm of the error vector [1]. The more similar a sample's prediction vector is to the label vector, the smaller EL2N score will a sample have. This means the sample is easier to learn and less informative for the training process. From this perspective, HEFS has identical properties as EL2N. Additionally, HEFS can be computed in HE more efficiently.
>
> (3) Batching data samples in ciphertext. Single Instruction-Multiple Data (SIMD) is a commonly used technique in HE. With SIMD, a single ciphertext has many slots, i.e., 32,768 slots in our setting. On the other hand, a single data sample can only occupy 768 slots in the transfer learning setting, and 784 slots for the MNIST dataset, 3,027 slots for the CIFAR-10 dataset. Packing multiple samples in one ciphertext is a commonly used strategy to reduce the number of ciphertexts as well as the number of HE operations [3,6].
>
> We will incorporate the reviewer's suggestions into the future versions for better clarity.
>
> [1] Paul, Mansheej, et al. Deep learning on a data diet: Finding important examples early in training.
>
> [2] Truong, Thao Nguyen, et al. KAKURENBO: adaptively hiding samples in deep neural network training.
>
> [3] Lee, Seewoo, et al. HETAL: efficient privacy-preserving transfer learning with homomorphic encryption.
>
> [4] Jawalkar, et al. Orca: FSS-based Secure Training and Inference with GPUs.
>
> [5] Viand, Alexander, et al. Verifiable fully homomorphic encryption.
>
> [6] Crockett, Eric. A low-depth homomorphic circuit for logistic regression model training.

---

> > ### Comment · Reviewer_98ig · 2024-08-13
> >
> > I have read the response. While it addresses some of my concerns, I still struggle with some fundamental issues such as how "the client [should determine] the pruning ratio" when "The client does not have adequate resource or knowledge for training".
> >
> > While it is true that the threat model can be strengthened by moving to verified outsourced computation techniques, that is a general problem with outsourced computation/FHE and wasn't my point, which was more that there's a lot of "side channel" information available to the outsourcing provider. What are the risks of this? Cryptographers have demonstrated that risks to data confidentiality can be enormous when even small channels exist. Verified computation can eliminate this by allowing the client to confirm that only desired computations are performed (during reported activities, but what about after or on the side?). If I give you ciphertext and you give me a ZKP that you did a certain computation using that ciphertext, that tells me zero information about what _else_ you did with that ciphertext. This is the problem. It's a very hard problem, and I don't mean that you have to solve it, but rather that you should acknowledge the limitation and declare it out of scope.

---

> ### Author Response · Authors · 2024-08-14
>
> We appreciate the reviewer's thorough reading of our response. We first explain the choice of pruning ratio in more detail. Similar to data pruning studies in plaintext, the pruning ratio is set as empirical values such as 0.5 or 0.3. Manually setting the pruning ratio does not require client-side training or expert knowledge. As shown in our previous responses and also data pruning studies in plaintext [1] (Section 3.3), a moderate pruning ratio can achieve close generalization performance compared to the full dataset. The client can choose an initial pruning ratio around 0.5, and adjust the initial pruning ratio according to the size of the training dataset, the quality of the data and the budget for the outsource training service. Moreover, the client can also adjust the pruning ratio during training according to the test accuracy. If you still have concerns about the pruning ratio, please let us know, and we will be happy to respond. We will clarify the choice of pruning ratio in our revision.
>
> We thank the reviewer for pointing out the side channel information during the threat model. We will clarify it and discuss the potential side channel risks and declare that securing the side channel risks is out of scope for this work more explicitly in the next-version manuscript.
>
> [1] Truong, Thao Nguyen, et al. KAKURENBO: adaptively hiding samples in deep neural network training.

---

### Author Rebuttal · Authors · 2024-08-07

We thank all reviewers for their constructive and insightful feedback.

We are glad that all reviewers unanimously agree that our encrypted data pruning methods are of great significance for improving private training. We appreciate the reviewers recognizing the novelty of our work as the first framework that enables data pruning in private training (FEVE, LFsM). Reviewers find our work and evaluation effective, and our presentation clear (98ig, FEVE, LFsM). The reviewers also find that our work successfully boosts the efficiency of private training without compromising either accuracy or privacy. This paper makes private training more practical in real-world applications.

Performing data pruning in the encrypted state is non-trivial. Directly adapting plaintext pruning methods is not viable. To start with, the importance score in the plaintext often involves complex nonlinear functions that are prohibitive in HE. Moreover, sorting samples are considered free in plaintext, but sorting in the encrypted state is expensive. Most importantly, the sample-wise pruning in plaintext leads to sparse ciphertexts and fails to effectively accelerate private training. We propose a series of HE-aware optimizations including HEFS, client-aided masking, and ciphertext-wise pruning. Our methods boost the efficiency of private training significantly without sacrificing accuracy.

We thank the reviewers for their interest in many aspects of our work.
**(1) Generality and scalability:** We have shown that the proposed encrypted data pruning methods are highly general and scalable for different datasets and pruning ratios.
**(2) Lightweight design:** We present a more detailed runtime breakdown to show the proposed methods are extremely efficient at runtime, taking as little as 0.4% to 2.5% of the total runtime of private training.
**(3) Analysis on Privacy and Efficiency:** We present an in-depth analysis of the privacy and efficiency of the proposed methods. We guarantee data privacy and model privacy simultaneously via HE throughout training and address security issues related to the pruning mask. We demonstrate the mechanism of using the proposed HEFS score to identify the most informative samples in the encrypted state. Meanwhile, we leverage HE-friendly scores, efficient sorting, and ciphertext-wise pruning techniques to construct lightweight yet effective strategies that significantly enhance the efficiency of private training.

---

### Decision · Program_Chairs · 2024-09-25

**Decision:**

Accept (poster)

**Comment:**

After discussion, all the reviewers provided positive feedback on this submission, therefore we recommend accepting it.